# Impact of acute decompensation on the prognosis of patients with hepatocellular carcinoma

Takayuki Kondo[1]*, Keisuke Koroki[1], Hiroaki Kanzaki[1], Kazufumi Kobayashi[1,2], Soichiro Kiyono[1], Masato Nakamura[1], Naoya Kanogawa[1], Tomoko Saito[1], Sadahisa Ogasawara[1,2], Yoshihiko Ooka[1], Shingo Nakamoto[1], Tetsuhiro Chiba[1], Makoto Arai[1,3], Jun Kato[1], Satoshi Kuboki[4], Masayuki Ohtsuka[4], Naoya Kato[1]

1 Department of Gastroenterology, Graduate School of Medicine, Chiba University, Chiba, Japan, 2 Translational Research and Development Center, Chiba University Hospital, Chiba, Japan, 3 Department of Medical Oncology, Graduate School of Medicine, Chiba University, Chiba, Japan, 4 Department of General Surgery, Graduate School of Medicine, Chiba University, Chiba, Japan

* takakondonaika@yahoo.co.jp

**Data Availability Statement:** All relevant data are within the paper.

**Funding:** The authors received no specific funding for this work.

## Abstract

### Background/Aims

Organ failure in patients with acute decompensation (AD) is a defining characteristic of acute-on-chronic liver failure (ACLF). However, the clinical features of AD during the long-term clinical course of hepatocellular carcinoma (HCC) are still poorly understood. This study aimed to clarify features and impact of AD/ACLF on the prognosis of patients after treatment for HCC.

### Methods

This retrospective study enrolled 556 consecutive patients who were initially diagnosed with HCC, and analyses were conducted taking into account HCC treatment type, HCC stage, and presence or absence of cirrhosis.

### Results

During follow-up, 299 patients with AD were hospitalized. AD occurrence is closely related to prognosis, regardless of the presence or absence of cirrhosis and HCC stage, and early-onset AD (within 90 days after HCC treatment) has negative impact on prognosis. In the intermediate-advanced–stage group, surgical resection had a positive impact on AD incidence post-treatment. After systemic therapy for HCC, renal impairment was the predictive factors for AD development. The 28/90-day mortality rate was higher among 41 cases (13.7%) with AD who exhibited ACLF as compared with cases without ACLF. AD without cirrhosis had similar ACLF incidence and short-term mortality, compared to AD with cirrhosis. The prognostic model using a decision-tree–based approach, which includes ACLF, bilirubin level, HCC progression, and MELD score is useful for predicting 90- or 28-day mortality after AD diagnosis.

**Competing interests:** The authors have declared that no competing interests exist.

**Abbreviations:** AASLD, American Association for the Study of Liver Disease; ACLF, acute-on-chronic liver failure; AD, acute decompensation; ALBI, albumin-bilirubin; APASL, Asian Pacific Association for the Study of the Liver; CART, classification and regression tree; CLIF-C AD, chronic liver failure-consortium acute decompensation; CLIF-OF, chronic liver failure-organ failure; EASL, European Association for the Study of the Liver; HCC, hepatocellular carcinoma; HE, hepatic encephalopathy; MELD, model for end-stage liver disease; ROC, receiver operating characteristics.

## Conclusions

Careful management of patients with HCC who are hospitalized with AD is necessary, considering ACLF, HCC progression, and liver function.

## Introduction

Hepatocellular carcinoma (HCC) remains one of the leading causes of cancer-related death worldwide.[1] Management of HCC should be based on proper assessment of disease severity, treatment, and surveillance [1–3]. Over the past few decades, despite the improvement of disease management, the outcomes of HCC remain unsatisfactory [1,4]. In patients with HCC, the prevalence of concomitant liver cirrhosis is greater than 80%, and the liver functional reserve is one of the critical factors affecting their prognosis [1–3].

The most common hospital presentation of patients with cirrhosis is acute decompensation (AD) with gastrointestinal bleeding, bacterial infection, acute onset of ascites, or hepatic encephalopathy (HE), alone or in combination [5–8]. The occurrence of hepatic and extrahepatic organ failure in patients with AD is indicative of acute-on-chronic liver failure (ACLF). ACLF occurs in approximately 30% of patients with acute AD and is associated with a 28-day mortality rate of roughly 30% [9–11].

However, the proposed definition of ACLF by the European Association for the Study of the Liver (EASL) and the American Association for the Study of Liver Disease (AASLD) has been validated only in patients with cirrhosis [10]. In contrast, the Asian Pacific Association for the Study of the Liver (APASL) definition of ACLF includes chronic hepatitis regardless of the presence of cirrhosis [12], because chronic liver disease is difficult to clearly distinguish from cirrhosis [13,14]. In addition, most of the previous reports that demonstrated the impact of AD in patients with cirrhosis excluded the HCC population [10]; thus, the influence of AD on the clinical outcome after treatment for HCC and the influence of treatment for HCC on the incidence of AD or ACLF remains unclear.

We hypothesized that AD occurrence is a crucial determinant of the outcomes after treatment for HCC, regardless of the presence of cirrhosis. To validate this hypothesis and the clinical features of AD after treatment for HCC, we conducted the present study to elucidate the occurrence of AD during the long-term clinical course, and the influence of AD on the prognosis of patients after treatment for HCC. In addition, we evaluated the ACLF incidence and impact of AD on short-term prognosis after treatment for HCC, comparing HCC with cirrhosis versus without cirrhosis.

## Patients and methods

### Patients

This retrospective study included data obtained from our institutional database between October 2011 and December 2016. We enrolled consecutive patients who were initially diagnosed with HCC and scheduled for treatment. Patients receiving maintenance dialysis or who received only best supportive care were excluded. The treatment strategy for HCC was discussed at a multidisciplinary meeting. After the advantages and side effects of various therapies and recommendations from the experts were explained, patients finalized the treatment strategy.

This study conformed to the principles of the Declaration of Helsinki and was approved by the Ethics Committee of Chiba University Graduate School of Medicine, and written informed

consent was waived because of the retrospective design. Informed consent was obtained in the form of an opt-out on the web-site.

## Definitions

HCC was diagnosed using the AASLD criteria (early-stage: single of any size or $\leq$ 3 nodules of $\leq$ 3 cm diameter; intermediate-stage: > 3 nodules of any size or 2–3 nodules of > 3 cm diameter; advanced-stage: any nodules with macrovascular invasion or extrahepatic spread). AD was defined according to the acute onset of ascites [15], HE [16], gastrointestinal/intra-abdominal hemorrhage [17], or bacterial infection [18], alone or in combination, requiring hospital treatment. ACLF was also defined with respect to organ failure according to the Chronic Liver Failure (CLIF)-Organ Failure (OF) score [9] and diagnosis required, in relation to known or unknown chronic liver disease. The CLIF-OF is scored as follows: grade 1 –(i) single kidney failure; (ii) single liver, coagulation, circulatory, or respiratory failure, and serum creatinine levels between 1.5 and 2 mg/dL and/or HE grade I or II; (iii) single cerebral failure (HE grade III or IV) associated with a serum creatinine between 1.5 and 2 mg/dL; grade 2, two organ failures; grade 3, three or more organ failures. Non-ACLF was defined as AD in patients who did not meet the criteria for ACLF diagnosis [9].

After we analyzed all non-ACLF patients at admission due to AD, the diagnosis of later-onset ACLF was reached. A patient was diagnosed with later-onset ACLF if the ACLF developed within 28 days after admission.

Evident cirrhosis was defined according to a combination of clinical signs and findings provided by laboratory tests, radiologic imaging, or liver biopsy. HE was assessed using the West Haven grading system [16]. The degree of ascites was defined according to international guidelines [15]: mild, ascites that was only detectable by ultrasound examination; moderate, ascites that caused moderate symmetrical distension of the abdomen; and severe, ascites that caused marked abdominal distension. Diagnosis of spontaneous bacterial peritonitis was confirmed when the ascitic neutrophil count was >250 cells/mm$^3$ with no intra-abdominal and surgically treatable source of sepsis [15].

## Statistical analysis

All data are expressed as the mean ± SD or as a percentage. Continuous variables were analyzed using Student's *t*-test or Mann–Whitney *U*-test, as appropriate. Categorical variables were analyzed using Fisher's exact test or chi-squared test, as appropriate. The multivariate analysis was assessed by logistic regression analysis. The cumulative survival rate was calculated using the Kaplan–Meier method. Risk factors for the development of AD were evaluated by Cox regression analysis. The best cut-off value was determined by receiver operating characteristics (ROC) analysis. Classification and regression tree (CART) analysis was performed using the R-powered data tool Exploratory (https::/exploratory.io) [19]. Statistical data were analyzed using SAS version 9.2 (SAS Institute, Cary, NC) and p < 0.05 was considered statistically significant.

## Results

### Patient characteristics

The study flowchart is depicted in Fig 1. Among 616 consecutive patients initially diagnosed with HCC during the study period, 556 were enrolled. Table 1 summarizes the clinical data for the study population. The median observation period was 36.7 months. Initially, 332 (59.7%) patients received potentially curative treatment (surgical resection or ablation). One patient

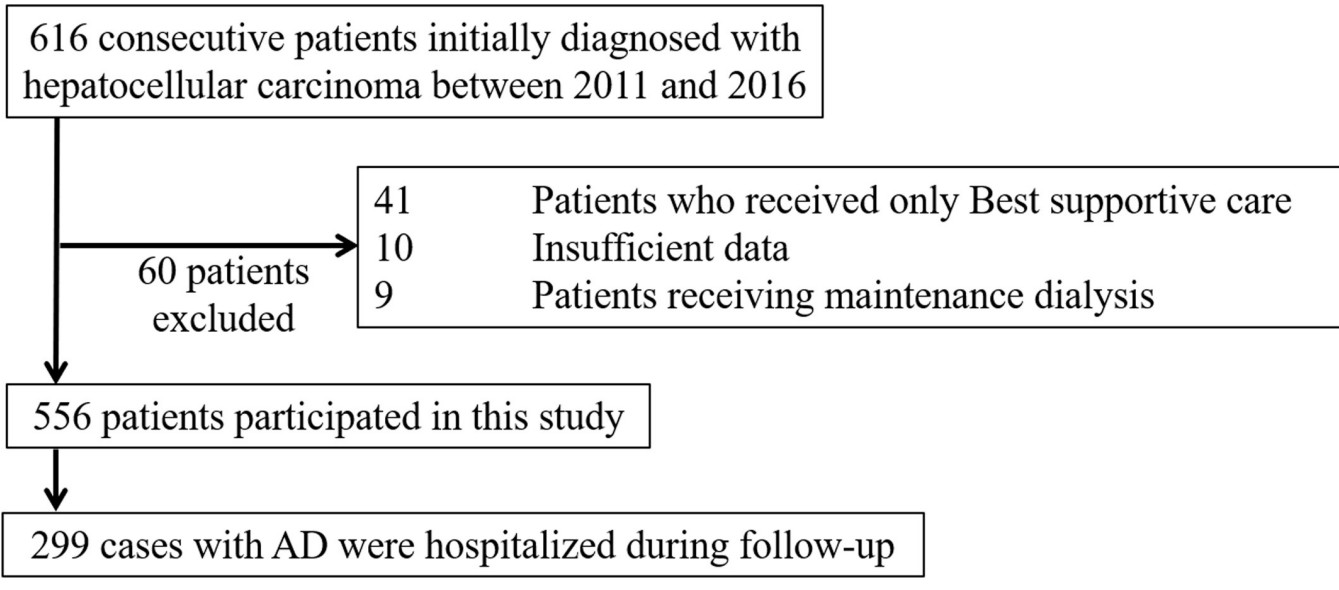

**Fig 1. Protocol diagram.**

received liver transplantation, and 223 patients died during the study period. The cumulative overall survival rates were 85.8% at 1 year and 66.8% at 3 years, and significantly lower in the intermediate-advanced–stage group (67.9% at 1 year, 36.1% at 3 years) than in the early-stage group (95.4% at 1 year, 83.8% at 3 years; p < 0.001).

## Development of AD

During follow-up, 299 cases with AD were hospitalized. The overall AD incidence rates were 15.5% at 1 year and 28.2% at 3 years.

Table 2 shows predictive factors for the development of AD according to univariate and multivariate analyses. The best cut-off value for predicting AD development as determined by

**Table 1. Patient characteristics.**

| | |
|---|---|
| Number of patients | 556 |
| HCC early/intermediate-advanced stage | 353/203 |
| Age (years) | 70 ± 9 |
| Sex (male/female) | 396/160 |
| Liver cirrhosis | 349 (62.7%) |
| Prior history of acute decompensation | 97 (17.5%) |
| Etiology (virus/alcohol /NASH/PBC/AIH/cryptogenic) | 339/71/34/8/4/100 |
| Albumin-bilirubin (ALBI) score | -2.42 ± 0.49 |
| Child-Pugh score | 6 ± 1 |
| Model for end-stage liver disease (MELD) score | 7 ± 2 |
| Alpha-fetoprotein >18 ng/ml* | 272 (48.9%) |
| Treatment for HCC (surgical resection/ablation/chemoembolization/systemic therapy) | 140/192/178/46 |

Data are presented as mean ± SD or number (%).

AIH, autoimmune hepatitis; HCC, hepatocellular carcinoma; NASH, nonalcoholic steatohepatitis; PBC, primary biliary cirrhosis.

*Median value of alpha-fetoprotein was 18 ng/ml.

**Table 2. Cox regression analyses of predictive factors for the development of acute decompensation.**

| | Univariate hazard ratio (95% confidence interval) | P value | Multivariate hazard ratio (95% confidence interval) | P value |
|---|---|---|---|---|
| Age | 0.991 (0.976–1.006) | 0.235 | - | |
| Male sex | 1.308 (0.945–1.811) | 0.105 | - | |
| Liver cirrhosis | 2.020 (1.459–2.796) | <0.001 | 1.843 (1.312–2.589) | <0.001 |
| Prior history of acute decompensation | 3.912 (2.821–5.426) | <0.001 | - | |
| Intermediate-advanced-stage HCC | 3.749 (2.791–5.034) | <0.001 | 2.970 (2.154–4.095) | <0.001 |
| Ascites | 3.166 (2.186–4.584) | <0.001 | - | |
| Treatment for HCC | | | | |
| Surgical resection | 0.459 (0.312–0.675) | <0.001 | - | |
| Ablation | 0.544 (0.399–0.742) | <0.001 | - | |
| Chemoembolization | 2.599 (1.942–3.478) | <0.001 | - | |
| Systemic therapy | 4.192 (2.566–6.848) | <0.001 | 1.815 (1.071–3.075) | 0.027 |
| Etiology | | | | |
| Virus related hepatitis | 0.532 (0.400–0.707) | <0.001 | 0.534 (0.396–0.719) | <0.001 |
| Laboratory data | | | | |
| Alanine aminotransferase (U/L) | 1.005 (1.002–1.008) | 0.003 | 1.003 (1.000–1.006) | 0.049 |
| Bilirubin (mg/dL) | 2.286 (1.801–2.902) | <0.001 | - | |
| Prothrombin time (international normalized ratio) | 5.960 (2.904–12.232) | <0.001 | - | |
| Albumin (g/dL) | 0.341 (0.254–0.456) | <0.001 | - | |
| Creatinine (mg/dL) | 1.444 (0.909–2.294) | 0.120 | - | |
| Sodium (mmol/L) | 0.941 (0.903–0.982) | 0.005 | - | |
| Platelets ($10^9$/L) | 1.008 (0.989–1.028) | 0.393 | - | |
| Alfa fetoprotein > 18 ng/ml* | 1.869 (1.401–2.493) | <0.001 | 1.362 (1.001–1.853) | 0.049 |
| Albumin-bilirubin (ALBI) score | 3.450 (2.542–4.682) | <0.001 | - | |
| Child-Pugh score | 2.042 (1.788–2.333) | <0.001 | 1.746 (1.508–2.021) | <0.001 |
| Model for end-stage liver disease (MELD) score | 1.173 (1.102–1.248) | <0.001 | - | |

HCC; hepatocellular carcinoma.

*Cut-off value was determined by receiver operating characteristics analysis.

ROC analysis was 18 ng/ml for AFP. In the multivariate analysis, presence of evident liver cirrhosis (p < 0.001), intermediate-advanced stage HCC (p < 0.001), higher alanine aminotransferase level (p = 0.049), higher AFP level (p = 0,049), higher Child-Pugh score (p < 0.001), and cause of cirrhosis other than virus-related hepatitis (p = ≤ 0.001) were significant predictive factors for AD development.

## Predictive factors for the development of AD stratified with a type of treatment strategy for HCC

As shown in Tables 3–6, Intermediate-advanced stage HCC was significantly associated with the development of AD after all treatment strategies other than systemic therapy. Cause of cirrhosis other than virus-related hepatitis was the significant predictive factor for the development of AD after potentially curative treatment (surgical resection group, p = 0.043; ablation group, p = 0.006). Prior history of AD (ablation group, p = 0.018), higher AFP level (ablation group, p = 0.011), higher ALBI score (ablation group, p = 0.003), higher Child-Pugh score (chemoembolization group, p < 0.001; systemic therapy group, p < 0.001), higher alanine aminotransferase level (chemoembolization group, p = 0.009), higher bilirubin level

**Table 3. Cox regression analyses of predictive factors for the development of acute decompensation in the surgical resection group.**

| | Univariate hazard ratio (95% confidence interval) | P value | Multivariate hazard ratio (95% confidence interval) | P value |
|---|---|---|---|---|
| Age | 1.022 (0.983–1.063) | 0.266 | - | |
| Male sex | 1.111 (0.496–2.491) | 0.798 | - | |
| Liver cirrhosis | 1.769 (0.871–3.591) | 0.114 | - | |
| Prior history of acute decompensation | 2.508 (0.337–18.634) | 0.369 | - | |
| Ascites | 0.000 (0.000-) | 0.990 | - | |
| Intermediate-advanced-stage HCC | 2.683 (1.235–5.828) | 0.013 | 2.733 (1.256–5.947) | 0.011 |
| Etiology | | | | |
| Virus related hepatitis | 0.475 (0.227–0.992) | 0.048 | 0.467 (0.223–0.997) | 0.043 |
| Laboratory data | | | | |
| Alanine aminotransferase (U/L) | 1.009 (0.999–1.019) | 0.069 | - | |
| Bilirubin (mg/dL) | 0.620 (0.210–1.824) | 0.385 | - | |
| Prothrombin time (international normalized ratio) | 1.963 (0.233–16.520) | 0.535 | - | |
| Albumin (g/dL) | 0.956 (0.425–2.153) | 0.914 | - | |
| Creatinine (mg/dL) | 2.650 (0.625–11.237) | 0.186 | - | |
| Sodium (mmol/L) | 0.978 (0.841–1.136) | 0.767 | - | |
| Platelets ($10^9$/L) | 1.011 (0.953–1.073) | 0.713 | - | |
| Alfa fetoprotein > 18 ng/ml* | 1.850 (0.910–3.760) | 0.089 | - | |
| Albumin-bilirubin (ALBI) score | 0.970 (0.386–2.436) | 0.949 | - | |
| Child-Pugh score | 0.921 (0.387–2.192) | 0.852 | - | |
| Model for end-stage liver disease (MELD) score | 1.142 (0.906–1.438) | 0.260 | - | |

HCC; hepatocellular carcinoma.

*Cut-off value was determined by receiver operating characteristics analysis.

(chemoembolization group, p = 0.014), and, higher creatinine level (systemic therapy group, p = 0.007) were the predictive factors for the development of AD after each type of treatment strategy for HCC.

## Predictive factors for the development of AD stratified with HCC stage

Tables 7 and 8 show the results of predictive factors for the development of AD stratifying HCC into early-stage HCC and intermediate-advanced-stage HCC. In early-stage HCC group, presence of evident liver cirrhosis (p < = 0.001), prior history of AD (p < 0.001), higher Child-Pugh score (p < 0.001), and cause of cirrhosis other than virus-related hepatitis (p = 0.001) were significant predictive factors for AD development. In intermediate-advanced-stage HCC group, advanced-stage HCC (p = 0.011), higher bilirubin level (p = 0.033), and higher Child-Pugh score (p < 0.001) were significant predictive factors for AD development, and surgical resection for HCC was negatively associated with AD development (p = 0.007).

## Impact of AD on prognosis

Cumulative overall survival was significantly lower in patients with the occurrence of AD than those without (without AD vs. with AD: 93.2% vs. 76.9% at 1 year/78.9% vs. 48.5% at 3 years, p < 0.001). When stratified AD into early-onset AD (within 90 days after treatment for HCC) and later-onset AD (after 90 days of treatment for HCC), early-onset AD negatively affected long-term survival, compared to later-onset AD (non-AD vs. later-onset AD vs. early-onset AD: 90.8% vs. 86.7% vs. 33.8% at 1 year/78.9% vs. 55.7% vs. 16.9% at 3 years, p < 0.001).

**Table 4. Cox regression analyses of predictive factors for the development of acute decompensation in the ablation group.**

| | Univariate hazard ratio (95% confidence interval) | P value | Multivariate hazard ratio (95% confidence interval) | P value |
|---|---|---|---|---|
| Age | 1.004 (0.972–1.038) | 0.805 | - | |
| Male sex | 0.941 (0.556–1.595) | 0.822 | - | |
| Liver cirrhosis | 3.535 (1.514–8.252) | 0.004 | - | |
| Prior history of acute decompensation | 3.443 (1.852–6.401) | <0.001 | 2.182 (1.143–4.166) | 0.018 |
| Ascites | 3.083 (1.455–6.530) | 0.003 | - | |
| Intermediate-advanced-stage HCC | 4.362 (1.334–14.264) | 0.015 | 4.418 (1.295–15.075) | 0.018 |
| Etiology | | | | |
| Virus related hepatitis | 0.438 (0.255–0.754) | 0.029 | 0.445 (0.249–0.795) | 0.006 |
| Laboratory data | | | | |
| Alanine aminotransferase (U/L) | 0.999 (0.991–1.006) | 0.737 | - | |
| Bilirubin (mg/dL) | 1.957 (1.111–3.450) | 0.021 | - | |
| Prothrombin time (international normalized ratio) | 4.128 (0.381–44.691) | 0.243 | - | |
| Albumin (g/dL) | 0.343 (0.190–0.618) | <0.001 | - | |
| Creatinine (mg/dL) | 1.253 (0.471–3.331) | 0.651 | - | |
| Sodium (mmol/L) | 0.976 (0.907–1.051) | 0.522 | - | |
| Platelets ($10^9$/L) | 0.934 (0.883–0.989) | 0.018 | - | |
| Alfa fetoprotein > 18 ng/ml* | 1.838 (1.097–3.080) | 0.021 | 2.020 (1.174–3.475) | 0.011 |
| Albumin-bilirubin (ALBI) score | 3.354 (1.810–6.215) | <0.001 | 2.606 (1.386–4.903) | 0.003 |
| Child-Pugh score | 1.812 (1.327–2.494) | <0.001 | - | |
| Model for end-stage liver disease (MELD) score | 1.065 (0.855–1.328) | 0.573 | - | |

HCC; hepatocellular carcinoma.

*Cut-off value was determined by receiver operating characteristics analysis.

Table 9 shows the prognostic factors. Multivariate analysis identified the following significant prognostic factors: the prior history of AD (p = 0.001), intermediate-advanced HCC (p < 0.001), receiving potentially curative treatment as the primary treatment (p = 0.001), the occurrence of AD during clinical course (p < 0.001), higher AFP level (p = 0.004), and higher Child-Pugh score (p < 0.001).

AD occurrence was associated with the poor prognosis after treatment for HCC, regardless of the presence or absence of cirrhosis (without AD vs with AD: absence of cirrhosis, 93.9% vs. 72.0% at 1 year/82.3% vs. 51.6% at 3years, p < 0.001; presence of cirrhosis, 88.2% vs. 78.5% at 1 year/77.6% vs. 47.3% at 3 years, p < 0.001) and the stage of HCC (without AD vs with AD: early-stage, 96.4% vs. 92.7% at 1 year/90.6% vs. 67.3% at 3years, p < 0.001; intermediate-advanced-stage, 76.0% vs. 59.7% at 1 year/45.3% vs. 26.7% at 3 years, p < 0.001).

## Development of ACLF

Among 299 cases with AD, 41 (13.7%) had ACLF at admission due to AD. In cases with and without ACLF (non-ACLF), the mortality rates were 68.3% versus 19.4% at 28 days after admission (p < 0.001) and 87.8% versus 41.5% at 90 days after admission (p < 0.001). The ACLF grade was associated with the short-term prognosis (28/90-day mortality: non-ACLF, 19.4%/41.5%; grade 1, 61.5%/92.0%; grades 2–3, 80.0%/86.7%, p < 0.001, respectively). Of the 258 non-ACLF cases at admission, 23 (8.9%) developed ACLF within 28 days (later-onset ACLF), and their 28- and 90-day mortality rates were significantly higher than patients who

**Table 5. Cox regression analyses of predictive factors for the development of acute decompensation in the chemoembolization group.**

| | Univariate hazard ratio (95% confidence interval) | P value | Multivariate hazard ratio (95% confidence interval) | P value |
|---|---|---|---|---|
| Age | 0.980 (0.958–1.001) | 0.067 | - | |
| Male sex | 1.146 (0.680–1.933) | 0.609 | - | |
| Liver cirrhosis | 1.414 (0.854–2.341) | 0.179 | - | |
| Prior history of acute decompensation | 2.301 (1.431–3701) | 0.001 | - | |
| Ascites | 1.622 (0.944–2.787) | 0.080 | - | |
| Intermediate-advanced-stage HCC | 2.260 (1.360–3.756) | 0.002 | 2.359 (1.405–3.962) | 0.001 |
| Etiology | | | | |
| Virus related hepatitis | 0.805 (0.523–1.238) | 0.323 | - | |
| Laboratory data | | | | |
| Alanine aminotransferase (U/L) | 1.005 (1.002–1.008) | 0.004 | 1.005 (1.001–1.009) | 0.009 |
| Bilirubin (mg/dL) | 2.451 (1.794–3.350) | <0.001 | 1.594 (1.097–2.315) | 0.014 |
| Prothrombin time (international normalized ratio) | 6.725 (2.155–20.990) | 0.001 | - | |
| Albumin (g/dL) | 0.358 (0.235–0.546) | <0.001 | - | |
| Creatinine (mg/dL) | 1.059 (0.582–1.928) | 0.852 | - | |
| Sodium (mmol/L) | 0.929 (0.873–0.988) | 0.019 | - | |
| Platelets ($10^9$/L) | 1.005 (0.983–1.028) | 0.638 | - | |
| Alfa fetoprotein > 18 ng/ml* | 1.250 (0.809–1.933) | 0.314 | - | |
| Albumin-bilirubin (ALBI) score | 3.266 (2.113–5.048) | <0.001 | - | |
| Child-Pugh score | 1.819 (1.506–2.197) | <0.001 | 1.681 (1.323–2.135) | <0.001 |
| Model for end-stage liver disease (MELD) score | 1.097 (1.015–1.187) | 0.020 | - | |

HCC; hepatocellular carcinoma.

*Cut-off value was determined by receiver operating characteristics analysis.

did not develop later-onset ACLF (15.7% vs. 56.5% at 28 days, p < 0.001; 37.0% vs. 87.0% at 90 days, p < 0.001).

Comparing AD with cirrhosis and those without, there is no significant difference in the ACLF incidence (with cirrhosis vs. without cirrhosis: 14.5% vs. 13.5%, p = 0.830), later-onset ACLF incidence (10.6% vs. 3.4%, p = 0.090), and 28-/90-day mortality (27.4% vs. 21.7% at 28 days, p = 0.348; 47.8% vs. 47.8% at 90 days, p = 1.000).

## Prognostic model for 28/90-day mortality in cases with AD

Table 10 lists the predictive factors for 28/90-day mortality, according to univariate analysis, among 299 cases with AD. In multivariate analysis, the significant predictive factors for 28-day mortality were AD with ALCF (odds ratio [OR], 6.510; 95% confidence interval [CI], 2.844–14.900; p < 0.001), HCC progression as a potential precipitating event (OR, 3.842; 95% CI, 1.846–7.998; p < 0.001), white blood cell level (OR, 1.079; 95% CI, 1.023–1.137; p = 0.005), bilirubin level (OR, 1.140; 95% CI, 1.056–1.231; p = 1.140), albumin level (OR, 0452; 95% CI, 0.237–0.862; p = 0.016), and bacterial infection as a complication defining AD (OR, 0.305; 95% CI, 0.144–0.647; p = 0.002). The significant predictive factors for 90-day mortality were presence of ascites (OR, 3.133; 95% CI, 1.531–6.410; p = 0.002), HCC progression as a potential precipitating event (OR, 5.561; 95% CI, 2.566–12.052; p < 0.001), chemoembolization as a potential precipitating event (OR, 0.228, 95% CI, 0.055–0.941, p = 0.041), white blood cell level (OR, 1.087; 95% CI, 1.026–1.152; p = 0.005), albumin level (OR, 0534; 95% CI, 0.296–0.965; p = 0.038), MELD score (OR, 1.143; 95% CI, 1.079–1.211; p < 0.001), and ascites as a

**Table 6. Cox regression analyses of predictive factors for the development of acute decompensation in the systemic therapy group.**

| | Univariate hazard ratio (95% confidence interval) | P value | Multivariate hazard ratio (95% confidence interval) | P value |
|---|---|---|---|---|
| Age | 0.987 (0.950–1.026) | 0.509 | - | |
| Male sex | - (0.000-) | 0.993 | - | |
| Liver cirrhosis | 2.181 (0.815–5.842) | 0.121 | - | |
| Prior history of acute decompensation | 2.551 (0.963–6.757) | 0.060 | - | |
| Ascites | 2.020 (0.780–5.234) | 0.148 | - | |
| Intermediate-advanced-stage HCC | 1.840 (0.242–13.996) | 0.556 | - | |
| Etiology | | | | |
| Virus related hepatitis | 0.497 (0.194–1.277) | 0.146 | - | |
| Laboratory data | | | | |
| Alanine aminotransferase (U/L) | 1.001 (0.988–1.014) | 0.886 | - | |
| Bilirubin (mg/dL) | 1.827 (1.100–3.035) | 0.020 | - | |
| Prothrombin time (international normalized ratio) | 3.866 (1.054–14.184) | 0.042 | - | |
| Albumin (g/dL) | 0.189 (0.065–0.549) | 0.002 | - | |
| Creatinine (mg/dL) | 20.510 (2.509–167.658) | 0.005 | 12.211 (1.973–75.576) | 0.007 |
| Sodium (mmol/L) | 1.035 (0.858–1.249) | 0.716 | - | |
| Platelets ($10^9$/L) | 1.028 (0.986–1.072) | 0.194 | - | |
| Alfa fetoprotein > 18 ng/ml* | 1.378 (0.315–6.036) | 0.670 | - | |
| Albumin-bilirubin (ALBI) score | 6.086 (2.079–17.816) | 0.001 | - | |
| Child-Pugh score | 2.297 (1.475–3.579) | <0.001 | 2.412 (1.488–3.909) | <0.001 |
| Model for end-stage liver disease (MELD) score | 1.309 (1.114–1.540) | 0.001 | - | |

HCC; hepatocellular carcinoma.

*Cut-off value was determined by receiver operating characteristics analysis.

complication defining AD (OR, 0.296; 95% CI 0.135–0.648; p = 0.002). To construct a prognostic model, the most relevant variables were chosen for CART analysis, and several trees were constructed using an exploratory strategy. The final selected tree-discriminated cases were classified according to the following 3 subpopulations with distinct prognoses: prognostic model for 28-day mortality, low risk (Non-ACLF and bilirubin level < 9 mg/dL; 28-day mortality, 16%), intermediate risk (Non-ACLF and bilirubin ≥ 9 mg/dL; 28-day mortality, 61%), and high risk (AD with ACLF; 28-day mortality, 68%); and prognostic model for 90-day mortality, low risk (MELD score < 18 and without HCC progression as a potential precipitating event; 90-day mortality, 30%), intermediate risk (MELD score < 18 and with HCC progression as a potential precipitating event; 90-day mortality, 68%) and high risk (MELD score ≥ 18; 90-day mortality, 86%).

## Discussion

This study determined the incidence of AD/ACLF and the risk factors for the occurrence of AD during the long-term clinical course after treatment for HCC, stratified with a type of HCC strategy and HCC stage. Our results suggested that AD occurrence was closely related with the prognosis after treatment for HCC, regardless of the presence or absence of cirrhosis and the stage of HCC, and early-onset AD had another impact on prognosis. Furthermore, this study showed that AD without cirrhosis had similar ACLF incidence and similar short-term mortality after admission due to AD, compared to AD with cirrhosis. The prognostic

**Table 7. Cox regression analyses of predictive factors for the development of acute decompensation in the early-stage group.**

| | Univariate hazard ratio (95% confidence interval) | P value | Multivariate hazard ratio (95% confidence interval) | P value |
|---|---|---|---|---|
| Age | 0.998 (0.976–1.020) | 0.853 | - | |
| Male sex | 0.845 (0.562–1.272) | 0.420 | - | |
| Liver cirrhosis | 2.902 (1.756–4.797) | <0.001 | 2.373 (1.402–4.016) | 0.001 |
| Prior history of acute decompensation | 3.945 (2.379–6.542) | <0.001 | 1.604 (1.260–2.043) | <0.001 |
| Ascites | 2.233 (1.158–4.307) | 0.017 | - | |
| Treatment for HCC | | | | |
| Surgical resection | 0.577 (0.356–0.937) | 0.026 | - | |
| Ablation | 1.019 (0.682–1.524) | 0.926 | - | |
| Chemoembolization | 2.058 (1.254–3.380) | 0.004 | - | |
| Systemic therapy | 6.432 (0.876–47.251) | 0.067 | - | |
| Etiology | | | | |
| Virus related hepatitis | 0.522 (0.350–0.779) | 0.002 | 0.504 (0.336–0.755) | 0.001 |
| Laboratory data | | | | |
| Alanine aminotransferase (U/L) | 1.002 (0.996–1.008) | 0.468 | - | |
| Bilirubin (mg/dL) | 2.240 (1.484–3.380) | <0.001 | - | |
| Prothrombin time (international normalized ratio) | 4.511 (1.078–18.872) | 0.039 | - | |
| Albumin (g/dL) | 0.350 (0.260–0.607) | <0.001 | - | |
| Creatinine (mg/dL) | 1.532 (0.701–3.351) | 0.285 | - | |
| Sodium (mmol/L) | 0.958 (0.901–1.019) | 0.175 | - | |
| Platelets ($10^9$/L) | 0.942 (0.908–0.978) | 0.002 | - | |
| Alfa fetoprotein > 18 ng/ml* | 1.515 (1.015–2.261) | 0.042 | - | |
| Albumin-bilirubin (ALBI) score | 3.379 (2.166–5.272) | <0.001 | - | |
| Child-Pugh score | 1.936 (1.550–2.419) | <0.001 | 1.604 (1.260–2.043) | <0.001 |
| Model for end-stage liver disease (MELD) score | 1.181 (1.054–1322) | 0.004 | - | |

HCC; hepatocellular carcinoma.

*Cut-off value was determined by receiver operating characteristics analysis.

model using a decision-tree–based approach, including ACLF, bilirubin level, HCC progression, and MELD score, was useful for predicting 90/28-day mortality after AD diagnosis.

The AD incidence (15.5%) at 1 year was higher in this study than that reported in the literature [20,21]. This finding is consistent with previous studies in which the incidence of decompensation was higher in patients with HCC than in those without HCC [22,23]. Decreased liver function is crucial for AD development. Some treatments for HCC [24,25], and the development of HCC itself, have a negative impact on liver function. Recently, surgery has been described as a trigger for ACLF in patients with AD because of the invasiveness of the procedure [26]. However, we showed that surgical resection had a positive impact on AD incidence in the intermediate-advanced–stage group. The underlying reason might be that surgery is likely to be selected for patients with better liver function and less severe HCC. In fact, patients who received surgery had lower Child-Pugh scores than those who underwent non-surgical treatment in this study (p < 0.001), and had no prior history of AD. This suggests that for patients with preserved liver function, surgery might be a treatment option even in cases with intermediate-advanced–stage HCC group. In addition, virus-infected patients showed a lower rate of AD incidence, especially in the group which received ablation or surgical resection, possibly because of improvements in HCV/HBV treatment. In fact, 57% of patients with virus-

**Table 8. Cox regression analyses of predictive factors for the development of acute decompensation in the intermediate-advanced-stage group.**

| | Univariate hazard ratio (95% confidence interval) | P value | Multivariate hazard ratio (95% confidence interval) | P value |
|---|---|---|---|---|
| Age | 0.993 (0.973–1.014) | 0.509 | - | |
| Male sex | 1.508 (0.804–2.829) | 0.201 | - | |
| Liver cirrhosis | 1.548 (1.000–2.397) | 0.050 | - | |
| Prior history of acute decompensation | 2.377 (1.535–3.682) | <0.001 | - | |
| Ascites | 2.617 (1.631–4.200) | <0.001 | - | |
| HCC advanced stage | 1.630 (1.080–2.459) | 0.020 | 1.780 (1.140–2.780) | 0.011 |
| Treatment for HCC | | | | |
| Surgical resection | 0.413 (0.214–0.797) | 0.008 | 0.378 (0.186–0.768) | 0.007 |
| Ablation | 0.883 (0.279–2.796) | 0.833 | - | |
| Chemoembolization | 1.198 (0.779–1.842) | 0.411 | - | |
| Systemic therapy | 1.950 (1.151–3.305) | 0.013 | - | |
| Etiology | | | | |
| Virus related hepatitis | 0.660 (0.439–0.992) | 0.046 | - | |
| Laboratory data | | | | |
| Alanine aminotransferase (U/L) | 1.003 (0.999–1.066) | 0.125 | - | |
| Bilirubin (mg/dL) | 2.191 (1.651–2.908) | <0.001 | 1.484 (1.032–2.134) | 0.033 |
| Prothrombin time (international normalized ratio) | 4.024 (1.790–9.046) | 0.001 | - | |
| Albumin (g/dL) | 0.406 (0.263–0.626) | <0.001 | - | |
| Creatinine (mg/dL) | 1.041 (0.591–1.835) | 0.889 | - | |
| Sodium (mmol/L) | 0.955 (0.899–1.014) | 0.135 | - | |
| Platelets ($10^9$/L) | 1.013 (0.993–1.033) | 0.208 | - | |
| Alfa fetoprotein > 18 ng/ml* | 1.393 (0.896–2.165) | 0.141 | - | |
| Albumin-bilirubin (ALBI) score | 2.922 (1.889–4.521) | <0.001 | - | |
| Child-Pugh score | 1.957 (1.631–2.347) | <0.001 | 1.592 (1.261–2.009) | <0.001 |
| Model for end-stage liver disease (MELD) score | 1.111 (1.029–1.200) | 0.007 | - | |

HCC; hepatocellular carcinoma.

*Cut-off value was determined by receiver operating characteristics analysis.

related HCC were treated with antivirals, and there was a significant difference in the development of AD between treated and non-treated patients (treated vs. non-treated: 5.4% vs. 17.6% at 1 year/12.7% vs. 42.4% at 3 years, p < 0.001). Apart from liver function, during systemic therapy for HCC, patients with renal impairment should be monitored carefully because of high risk of AD development. In terms of long-term mortality, cirrhosis was not a significant prognostic factor in our study. However, prior history of AD and AD occurrence were associated with prognosis. In addition, early-onset AD after HCC treatment had negative impact on prognosis, compared to later-onset AD. These results indicate that AD, rather than cirrhosis, represents poor liver function and has negative impact on prognosis.

Notably, ACLF incidence in our study was relatively low, reaching 14% [8,27], but this may be because patients with HCC undergo stricter follow-up and are managed more carefully. Another possible reason is that ACLF incidence was around 21% when later-onset ACLF was included in the analysis, comparable to previous reports [10,27,28]. Moreover, even when cases of later-onset ACLF were excluded, the short-term mortality rate of AD without ACLF was very high compared with the reported short-term mortality rates for AD and ACLF [10,13]. One of the probable causes of the high mortality rate of AD is cancer cachexia caused

**Table 9. Cox regression analyses of prognostic factors.**

| | Univariate hazard ratio (95% confidence interval) | *P* value | Multivariate hazard ratio (95% confidence interval) | *P* value |
|---|---|---|---|---|
| Age | 0.984 (0.971–0.998) | 0.021 | - | |
| Male sex | 1.623 (1.174–2.244) | 0.003 | - | |
| Liver cirrhosis | 1.597 (1.190–2.143) | 0.002 | - | |
| Prior history of acute decompensation | 4.270 (3.175–5.744) | <0.001 | 1.839 (1.294–2.613) | 0.001 |
| Ascites | 4.611 (3.555–6.336) | <0.001 | - | |
| Intermediate-advanced-stage HCC | 4.851 (3.672–6.408) | <0.001 | 2.547 (1.792–3.620) | <0.001 |
| Acute decompensation during clinical course | 3.985 (2.950–5.141) | <0.001 | 2.371 (1.769–3.177) | <0.001 |
| Treatment for HCC | | | | |
| Potentially curative treatment | 0.217 (0.164–0.287) | <0.001 | 0.546 (0.380–0.785) | 0.001 |
| Etiology | | | | |
| Virus related hepatitis | 0.635 (0.485–0.831) | 0.001 | - | |
| Laboratory data | | | | |
| Alanine aminotransferase (U/L) | 1.005 (1.002–1.008) | <0.001 | - | |
| Bilirubin (mg/dL) | 1.747 (1.378–2.216) | <0.001 | - | |
| Prothrombin time (international normalized ratio) | 4.996 (2.425–10.295) | <0.001 | - | |
| Albumin (g/dL) | 0.373 (0.284–0.490) | <0.001 | - | |
| Creatinine (mg/dL) | 1.390 (0.883–2187) | 0.155 | - | |
| Sodium (mmol/L) | 0.944 (0.908–0.983) | 0.005 | - | |
| Platelets ($10^9$/L) | 1.022 (1.006–1.039) | 0.008 | - | |
| Alfa fetoprotein > 18 ng/ml* | 2.228 (1.691–2.936) | <0.001 | 1.524 (1.147–2.024) | 0.004 |
| Albumin-bilirubin (ALBI) score | 2.907 (2.193–3.855) | <0.001 | - | |
| Child-Pugh score | 1.997 (1.763–2.262) | <0.001 | 1.449 (1.242–1.690) | <0.001 |
| Model for end-stage liver disease (MELD) score | 1.130 (1.058–1.207) | <0.001 | - | |

HCC; hepatocellular carcinoma.

*Cut-off value was determined by receiver operating characteristics analysis.

by HCC, considering the result of high short-term mortality in AD caused by HCC progression. Another possible explanation is that cancer-related inflammation may be strongly related to the high short-term mortality rate of AD in patients with HCC, which might be supported by the result of poor short-term prognosis in AD with elevated white blood cell level. Inflammation caused by viral hepatitis, alcoholic hepatitis, and NASH leads to hepatocyte death and the repetitive division and proliferation of hepatocytes (so-called compensatory proliferation), which results in genetic mutations caused by DNA replication errors and the development of fibrosis and HCC [29]. Aside from the background inflammatory state of the liver, malignant transformation of cells induces an aberrant functional response to produce proinflammatory mediators in the tumor microenvironment, even without prior causative inflammation, and triggers the expression of inflammatory mediators such as cytokines and chemokines that amplify the inflammatory symptoms [29–31]. In contrast, bacterial infection as a complication defining AD had positive impact on short-term prognosis. This may be because patients with HCC tend to be treated promptly with antibiotics, which lead to early control of bacterial infection.

As to whether or not to include chronic hepatitis cases in the ACLF diagnosis, our study showed that cirrhosis was associated with AD development, but the short-term mortality was not significantly different between chronic hepatitis and cirrhosis cases. The results were also

**Table 10. Patient characteristics stratified by survivors or non-survivors at day 28 and day 90 in cases with acute decompensation.**

| | 28-day | | | 90-day | | |
|---|---|---|---|---|---|---|
| | Survivors (N = 221) | Non-survivors (N = 78) | *P* value | Survivors (N = 156) | Non-survivors (N = 143) | *P* value |
| HCC intermediate-advanced stage | 91 (41.2%) | 43 (55.1%) | 0.003 | 61 (39.1%) | 73 (51.1%) | 0.038 |
| ACLF | 13 (5.9%) | 28 (35.9%) | <0.001 | 5 (3.2%) | 36 (25.2%) | <0.001 |
| Age (years) | 72 ± 8 | 68 ± 10 | 0.003 | 72 ± 8 | 70 ± 9 | 0.008 |
| Male | 155 (70.1%) | 61 (78.2%) | 0.171 | 105 (67.3%) | 111 (77.6%) | 0.047 |
| Liver cirrhosis | 167 (75.6%) | 63 (80.8%) | 0.348 | 120 (76.9%) | 110 (76.9%) | 1.000 |
| Prior history of acute decompensation | 62 (28.1%) | 28 (35.9%) | 0.194 | 45 (28.9%) | 45 (31.5%) | 0.621 |
| Ascites | 138 (62.4%) | 66 (84.6%) | <0.001 | 87 (55.8%) | 117 (81.8%) | <0.001 |
| Etiology | | | | | | |
| Virus related hepatits | 100 (45.3%) | 35 (44.9%) | 0.954 | 74 (47.4%) | 61 (42.7%) | 0.407 |
| Complications defining AD | | | | | | |
| Ascites | 48 (21.7%) | 13 (16.7%) | 0.341 | 39 (25.0%) | 22 (15.4%) | 0.039 |
| Hepatic encephalopathy | 26 (11.8%) | 18 (23.1%) | 0.015 | 18 (11.5%) | 26 (18.2%) | 0.105 |
| Hemorrhage | 57 (25.8%) | 28 (35.9%) | 0.089 | 41 (26.3%) | 44 (30.8%) | 0.390 |
| Bacterial infection | 90 (40.7%) | 19 (24.4%) | 0.010 | 58 (37.2%) | 51 (35.7%) | 0.786 |
| HCC related potential precipitating events | | | | | | |
| HCC progression | 29 (13.1%) | 29 (37.2%) | <0.001 | 13 (8.3%) | 45 (31.5%) | <0.001 |
| Systemic chemotherapy | 6 (2.7%) | 3 (3.9%) | 0.615 | 5 (3.2%) | 4 (2.8%) | 0.837 |
| Chemoembolization | 17 (7.7%) | 2 (2.6%) | 0.110 | 15 (9.6%) | 4 (2.8%) | 0.016 |
| Surgical resection/ablation | 5 (2.3%) | 2 (2.6%) | 0.880 | 3 (1.9%) | 4 (2.8%) | 0.618 |
| Laboratory data | | | | | | |
| Alanine aminotransferase (U/L) | 61 ± 122 | 97 ± 130 | 0.034 | 66 ± 142 | 76 ± 104 | 0.457 |
| White blood cell (x$10^9$/L) | 7.9 ± 5.0 | 10.6 ± 7.3 | 0.004 | 7.4 ± 5.0 | 10.0 ± 6.3 | <0.001 |
| Bilirubin (mg/dL) | 2.6 ± 2.9 | 5.6 ± 6.0 | <0.001 | 2.3 ± 2.5 | 4.6 ± 5.1 | <0.001 |
| Prothrombin time (international normalized ratio) | 1.24 ± 0.30 | 1.39 ± 0.34 | <0.001 | 1.22 ± 0.30 | 1.35 ± 0.32 | 0.001 |
| Albumin (g/dL) | 2.8 ± 0.5 | 2.5 ± 0.5 | <0.001 | 2.9 ± 0.5 | 2.6 ± 0.5 | <0.001 |
| Creatinine (mg/dL) | 1.04 ± 0.53 | 1.52 ± 0.84 | <0.001 | 0.96 ± 0.45 | 1.39 ± 0.77 | <0.001 |
| Sodium (mmol/L) | 135 ± 5 | 134 ± 7 | 0.294 | 136 ± 4 | 133 ± 6 | <0.001 |
| Platelets ($10^9$/L) | 137 ± 99 | 163 ± 120 | 0.092 | 131 ± 99 | 158 ± 111 | 0.026 |
| CLIF-C AD score | 53.8 ± 8.7 | 60.5 ± 8.2 | <0.001 | 52.0 ± 8.6 | 59.4 ± 8.0 | <0.001 |
| ALBI score | -1.39 ± 0.54 | -0.94 ± 0.55 | <0.001 | -1.47 ± 0.51 | -1.05 ± 0.57 | <0.001 |
| Child-Pugh score | 8 ± 2 | 10 ± 2 | < 0.001 | 8 ± 2 | 9 ± 2 | <0.001 |
| MELD score | 11 ± 5 | 18 ± 7 | <0.001 | 10 ± 4 | 16 ± 7 | <0.001 |

Data are expressed as mean ± SD or number (%).

ACLF, acute-on-chronic liver failure; AD, acute decompensation; ALBI, albumin-bilirubin; CLIF-C AD, chronic liver failure-consortium acute decompensation; HCC, hepatocellular carcinoma; MELD, model for end-stage liver disease.

similar in cases with a history of decompensation, consistent with previous reports in which cases without a history of AD had higher levels of inflammatory mediators than those with a history of AD [10]. Further study is needed to determine whether chronic hepatitis cases should be included in the ACLF diagnosis. However, even in chronic hepatitis cases, careful management of AD in patients after treatment for HCC is necessary.

Similar to the conclusion of previous studies [32,33], all of the scoring systems used in the present study, including the Child–Pugh, ALBI, CLIF-C AD, and MELD scores, correlated

well with the prognosis in the present study. In addition, given the high short-term mortality rate, liver transplantation might be considered in cases with HCC who meet MELD score ≥ 18 and Milan criteria.

Our study had several limitations. First, the data were retrospectively analyzed. Second, we excluded patients undergoing dialysis because of the problem of AD events in dialysis cases being diagnosed as ACLF. Therefore, the concept of ACLF in patients undergoing dialysis needs to be re-examined in a large cohort.

In conclusion, the occurrence of AD in patients with HCC correlates with a high mortality rate regardless of the stage of cirrhosis or liver cancer. In addition, patients with HCC who are hospitalized with AD should be carefully managed even if they have not reached the ACLF stage. Moreover, patients with impaired renal function who receive systemic therapy should be monitored carefully because of high incidence of AD. Lastly, a basic model using a simple CART algorithm is useful for estimating the prognosis for short-term mortality.

## Author Contributions

**Conceptualization:** Takayuki Kondo.

**Data curation:** Takayuki Kondo, Keisuke Koroki, Hiroaki Kanzaki, Kazufumi Kobayashi, Soichiro Kiyono, Masato Nakamura, Naoya Kanogawa, Tomoko Saito, Sadahisa Ogasawara, Yoshihiko Ooka, Shingo Nakamoto, Tetsuhiro Chiba, Makoto Arai, Jun Kato, Satoshi Kuboki, Masayuki Ohtsuka.

**Formal analysis:** Takayuki Kondo, Sadahisa Ogasawara, Naoya Kato.

**Investigation:** Takayuki Kondo, Sadahisa Ogasawara, Naoya Kato.

**Project administration:** Takayuki Kondo.

**Supervision:** Sadahisa Ogasawara, Masayuki Ohtsuka, Naoya Kato.

**Writing – original draft:** Takayuki Kondo.

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
