## [Decision Letter · Decision Letter 0]

6 Aug 2021

PONE-D-21-23680

Impact of acute decompensation on the prognosis of patients with hepatocellular carcinoma

PLOS ONE

Dear Dr. Kondo,

Thank you for submitting your manuscript to PLOS ONE. After careful consideration, we feel that it has merit but does not fully meet PLOS ONE’s publication criteria as it currently stands. Therefore, we invite you to submit a revised version of the manuscript that addresses the points raised during the review process.

While both reviewers appreciated the usefulness of your dataset, they both have difficulties with the structure of the manuscript, a clear definition of hypothesis etc. This issues need to be improved before the manuscript can be considered further.

We look forward to receiving your revised manuscript.

Kind regards,

Pavel Strnad

Academic Editor

PLOS ONE

Journal Requirements:

Reviewers' comments:

Reviewer's Responses to Questions

**Comments to the Author**

1. Is the manuscript technically sound, and do the data support the conclusions?

Reviewer #1: Partly

Reviewer #2: Partly

2. Has the statistical analysis been performed appropriately and rigorously? 

Reviewer #1: Yes

Reviewer #2: Yes

3. Have the authors made all data underlying the findings in their manuscript fully available?

Reviewer #1: Yes

Reviewer #2: Yes

4. Is the manuscript presented in an intelligible fashion and written in standard English?

Reviewer #1: Yes

Reviewer #2: No

5. Review Comments to the Author

Reviewer #1: I have read the article of Kondo et al “Impact of acute decompensation on the prognosis of patients with hepatocellular carcinoma” with great interest.

It describes an important topic on prognosis of HCC patients with HCC and the impact of acute decompensation. The aim of the study was to evaluate features and the impact of AD on the prognosis of HCC patients. The authors conducted a large retrospective study with 563 patients diagnosed with HCC, stratified them into early- and intermediate/advance group and measured occurrence of AD and ACLF episodes. The authors use internationally approved definitions and statistics are adequate.

The major comment I have is that the hypothesis/aim is somewhat confusing. On one hand they state that most studies that evaluated impact of AD in cirrhosis excluded HCC patients, and therefore the impact of AD on HCC patients is unknown. On the other hand, the authors hypothesize that AD occurrence is a crucial determinant of outcomes in HCC patients, regardless of cirrhosis. This are two important different questions. The first one requires a comparison between HCC cirrhotic patients and cirrhotic patients without HCC, and compare the occurrence of AD and the impact on survival. The second question requires a comparison between HCC patients with and without cirrhosis, and evaluate the occurrence on AD and prognosis.

By performing the analyses on the whole group (including HCC with cirrhosis and without cirrhosis) but without the comparison to a group without HCC, it is difficult to interpret these results. The main finding is that AD/ACLF occurrence is related to prognosis, but this is also the case in cirrhotic patients without HCC.

In addition, different treatment strategies were used, with potential different impacts on possible occurrence of AD. How did the authors correct for this effect, it seems important to me to separate the early complications from the potential beneficial effect in the long term when evaluating the exact impact of HCC and treatment on AD occurrence? I can imagine that an early complication presenting as AD has another impact on prognosis than an AD episode occurring years after the treatment. Moreover, the characteristics of patients that are treated with chemoembolization are completely different that patients with surgical resection or systemic therapy, it is difficult to compare these patient groups directly. I can imagine that an episode of AD impact survival differently in systemic therapy (with a HCC still present in the liver) and after resection (without an HCC). Ideally, these groups should be analyzed separately.

Finally, I’m missing AD occurrence (as a total group) in the prognostic models, given the aim of the study this would also be interesting to evaluate. Why did the authors stratify for early and advanced stage in prognostic modelling? It is also possible to include this variable as a predictive in the model, this would also results in larger groups which increases the power of the analysis.

In conclusion, while the authors have a very interesting dataset, the current analyses do not adequately answer the aims of the study. My suggestion is to remove most of the underpowered subgroup analyses and to focus on a main question that they want to answer, for example occurrence of AD after HCC treatment and impact on prognosis (but then results should be presented separately per treatment and take into account the different HCC populations undergoing these treatments). In the current form it is difficult to place these results in clinical context.

Minor comments:

- Why did the authors exclude Child-pugh C patients? Or patients with performance status of 2-3?

- Was liver biopsy known in every patient? The methods sections suggests this but it is not standard to biopsy every cirrhotic patient

- The numbers of the ACLF group are too low to perform prognostic modelling.

Reviewer #2: Kondo et al. performed a retrospective study to evaluate the impact of AD and ACLF on outcome in patients with HCC. Authors were trying to identify independent factors impacting on prognosis in this cohort.

Although this topic is interesting it is a difficult to read paper mostly due to the sequence results are presented. Different types of analysis (frequency of events, prognosis, univariate analysis, multivariate analysis) and distinct subgroups are put together into paragraphs which makes it difficult to comprehend the final messages.

General aspects:

I suggest, and this is my understanding of the manuscript, that authors make sure to address the following aspects and to put them in a more logical order:

• What is the relevance of AD and ACLF in cirrhosis with HCC (frequency and impact on prognosis)

• Is HCC and/or the stage of liver disease a factor impacting on the frequency of AD/ACLF (e.g. frequency and impact of HCC treatment related AD/ACLF, comparison with non-treatment related AD/ACLF; how many HCC related and how many AD/ACLF related deaths etc).

• Are there further independent factors impacting on patients’ outcome?

More specific aspects:

I am also missing the justification for dividing the groups upfront into early and late stage HCC. Although it is logical from the oncological point of view this paper is trying to elaborate on the decompensated cirrhosis/ACLF, which is meant to be biologically different. Authors should explain why they chose to do this subgroup analysis. I understand the results as if there are many data suggesting the severity of liver disease rather than the HCC itself as the main determinant for risk of AD/ACLF and subsequent outcome.

Why did authors include patients without cirrhosis? EASL CLIF ACLF definition requires the presence of liver cirrhosis.

At baseline, did all patients included have compensated cirrhosis?

Please explain the rational for excluding Child-C patients?

How many patients were transplanted during follow up? I understood that eligibility for transplant was no exclusion criterion.

Authors use the terms admission and hospitalisation – is that meant to be a surrogate for admission due to AD/ACLF? It is a bit confusing as admissions in general are not always due to AD/ACLF.

What is meant by curative treatment – intended to be or ultimately curative?

How many patients developed AD/ACLF as the consequence of HCC therapy? Are there factors impacting on AD/ACLF after HCC therapy?

Were all patients with viral hepatitis treated with antivirals? Any information on viral load etc? Were there differences between treated and non-treated patients?

There are too many variables in the regression analysis. For treatment and aetiology it may be useful to have only a binary variable (categorical), e.g. treatment/no treatment or viral hepatitis/others

Table 5 – Ascites, HE etc…. are no precipitating events, but complications of cirrhosis defining decompensation.

6. PLOS authors have the option to publish the peer review history of their article (what does this mean?). If published, this will include your full peer review and any attached files.

Reviewer #1: No

Reviewer #2: No

---

## [Author Response · Author response to Decision Letter 0]

23 Oct 2021

PONE-D-21-23680

Impact of acute decompensation on the prognosis of patients with hepatocellular carcinoma

PLOS ONE

To the Reviewers

Thank you for your suggestions and critical review.

Reviewer #1: I have read the article of Kondo et al “Impact of acute decompensation on the prognosis of patients with hepatocellular carcinoma” with great interest.

It describes an important topic on prognosis of HCC patients with HCC and the impact of acute decompensation. The aim of the study was to evaluate features and the impact of AD on the prognosis of HCC patients. The authors conducted a large retrospective study with 563 patients diagnosed with HCC, stratified them into early- and intermediate/advance group and measured occurrence of AD and ACLF episodes. The authors use internationally approved definitions and statistics are adequate.

The major comment I have is that the hypothesis/aim is somewhat confusing. On one hand they state that most studies that evaluated impact of AD in cirrhosis excluded HCC patients, and therefore the impact of AD on HCC patients is unknown. On the other hand, the authors hypothesize that AD occurrence is a crucial determinant of outcomes in HCC patients, regardless of cirrhosis. This are two important different questions. The first one requires a comparison between HCC cirrhotic patients and cirrhotic patients without HCC, and compare the occurrence of AD and the impact on survival. The second question requires a comparison between HCC patients with and without cirrhosis, and evaluate the occurrence on AD and prognosis.

By performing the analyses on the whole group (including HCC with cirrhosis and without cirrhosis) but without the comparison to a group without HCC, it is difficult to interpret these results. The main finding is that AD/ACLF occurrence is related to prognosis, but this is also the case in cirrhotic patients without HCC.

In addition, different treatment strategies were used, with potential different impacts on possible occurrence of AD. How did the authors correct for this effect, it seems important to me to separate the early complications from the potential beneficial effect in the long term when evaluating the exact impact of HCC and treatment on AD occurrence? I can imagine that an early complication presenting as AD has another impact on prognosis than an AD episode occurring years after the treatment. Moreover, the characteristics of patients that are treated with chemoembolization are completely different that patients with surgical resection or systemic therapy, it is difficult to compare these patient groups directly. I can imagine that an episode of AD impact survival differently in systemic therapy (with a HCC still present in the liver) and after resection (without an HCC). Ideally, these groups should be analyzed separately.

Finally, I’m missing AD occurrence (as a total group) in the prognostic models, given the aim of the study this would also be interesting to evaluate. Why did the authors stratify for early and advanced stage in prognostic modelling? It is also possible to include this variable as a predictive in the model, this would also results in larger groups which increases the power of the analysis.

In conclusion, while the authors have a very interesting dataset, the current analyses do not adequately answer the aims of the study. My suggestion is to remove most of the underpowered subgroup analyses and to focus on a main question that they want to answer, for example occurrence of AD after HCC treatment and impact on prognosis (but then results should be presented separately per treatment and take into account the different HCC populations undergoing these treatments). In the current form it is difficult to place these results in clinical context.

Reply: The authors agree with the points raised by the Reviewer. Following the Reviewer’s suggestion, our study focused on occurrence of AD after HCC treatment and impact on prognosis, and added a comparison between HCC patients with and without cirrhosis. Therefore, we have changed the relevant sentences in the Abstract section (marked-up copy; page 3, lines 5-6, 8-12, 14-16, 18-23, 25; page 4 line 1), the Introduction section (marked-up copy; page 5, lines 21-25; page 6, lines 1, 3-7), the Patients and Methods section (marked-up copy; page 7, lines 5-8, 20-22), the Result section (marked-up copy; page 10, lines 4-5, 7-8, 10-13, 16-24; page 11, lines 2-4, 6-9, 11-25; page 12, lines 1-25; page 13, lines 1-5, 8-9, 14-20; page 14, lines 1-4, 8-25; page 15, lines 1-25; page 16, lines 1-25), the Discussion section (marked-up copy; page 17; lines 3-15; page 18, lines 3-4, 7-9, 17-19, 21-22; page 19, lines 5-8, 9-10, 15, 17-25; page 20, lines 1-3, 8-14), Figure 1, and Tables 2-10. 

Minor comments:

- Why did the authors exclude Child-pugh C patients? Or patients with performance status of 2-3?

Reply: Thank you for your comments. According to your major comment, we had changed the aim to clarify the impact of AD after treatment for HCC. Therefore, we changed the exclusion criteria as follows: “Patients receiving maintenance dialysis or who received only best supportive care were excluded.” (marked-up copy; page 7, lines 8-11). In addition, we changed the relative sentences in the Abstract section (marked-up copy; page 3, lines 7, 11, 17), the Results section (marked-up copy; page 10, lines 4, 7, 12-13, 16-18; page 13, lines 8-11, 13-14, 20, 23-24; page 14 line 8), and the Discussion section (marked-up copy; page 18, line 12).

- Was liver biopsy known in every patient? The methods sections suggests this but it is not standard to biopsy every cirrhotic patient

Reply: We apologize for the unclear message in the original manuscript. We modified the relevant sentence (marked-up copy; page 8, line 10).

- The numbers of the ACLF group are too low to perform prognostic modelling.

Reply: According to your comment, we excluded the ACLF group from performing prognostic modeling (marked-up copy; page 14, lines 6-25; page 15, lines 1-12).

Reviewer #2: Kondo et al. performed a retrospective study to evaluate the impact of AD and ACLF on outcome in patients with HCC. Authors were trying to identify independent factors impacting on prognosis in this cohort.

Although this topic is interesting it is a difficult to read paper mostly due to the sequence results are presented. Different types of analysis (frequency of events, prognosis, univariate analysis, multivariate analysis) and distinct subgroups are put together into paragraphs which makes it difficult to comprehend the final messages.

General aspects:

I suggest, and this is my understanding of the manuscript, that authors make sure to address the following aspects and to put them in a more logical order:

• What is the relevance of AD and ACLF in cirrhosis with HCC (frequency and impact on prognosis)

Reply: According to your suggestion, we changed paragraph structure in the Result section as follows: #1 Patient characteristics, #2 Development of AD, #3 Predictive factors for the development of AD stratified with a type of treatment strategy for HCC, #4 Predictive factors for the development of AD stratified with HCC stage, #5 Impact of AD on prognosis, #6 Development of ACLF, #7 Prognostic model for 28/90-day mortality in cases with AD.

• Is HCC and/or the stage of liver disease a factor impacting on the frequency of AD/ACLF (e.g. frequency and impact of HCC treatment related AD/ACLF, comparison with non-treatment related AD/ACLF; how many HCC related and how many AD/ACLF related deaths etc).

Reply: We modified the analyses of the predictive factor of AD development and prognostic factor for short-term mortality after AD diagnosis, taking into account HCC treatment type and HCC stage. Therefore, we changed the relevant sentences in the Results section (marked-up copy; page 11, lines 11-25; page 12, lines 1-25; page 13, lines1-5; page 14, lines 6-25; page 15, lines 1-12) and Tables 2-10. In terms of ACLF, according to Reviewer 1’s suggestion, we excluded the ACLF group to perform prognostic modeling. During the study period, 223 patients died (marked-up copy; page 10, lines 9-10). However, it is difficult to distinguish HCC related death from AD/ACLF related death, especially in patients with advanced-stage HCC.

• Are there further independent factors impacting on patients’ outcome?

Reply: Thank you for your comment. We added Table 9 as analyses for prognostic factors and the sentences in the Result section (marked-up copy; page 12, lines 19-23).

More specific aspects:

I am also missing the justification for dividing the groups upfront into early and late stage HCC. Although it is logical from the oncological point of view this paper is trying to elaborate on the decompensated cirrhosis/ACLF, which is meant to be biologically different. Authors should explain why they chose to do this subgroup analysis. I understand the results as if there are many data suggesting the severity of liver disease rather than the HCC itself as the main determinant for risk of AD/ACLF and subsequent outcome.

Reply: Thank you for your comment. Please also see the response to Reviewer 1’s major comment. We changed the analyses to include HCC stage as a predictive factor and conducted subgroup analyses stratified HCC stage (marked-up copy; Tables 2-10).

Why did authors include patients without cirrhosis? EASL CLIF ACLF definition requires the presence of liver cirrhosis.

Reply: Thank you for your comment. We apologize for the unclear message in the original manuscript. We hypothesize that AD is a crucial event regardless of the presence of cirrhosis. Therefore, we added a comparison between HCC patients with and without cirrhosis to clarify the hypothesis (marked-up copy; page 6, lines 4-7; page 14, lines 1-4). The results showed that AD without cirrhosis had similar ACLF incidence and similar short-term mortality after admission due to AD, compared to AD without cirrhosis.

At baseline, did all patients included have compensated cirrhosis?

Please explain the rational for excluding Child-C patients?

Reply: Thank you for your comment. According to Reviewer 1’s comment, we had changed the aim to clarify the impact of AD after treatment for HCC, so we changed the exclusion criteria as follows: “Patients receiving maintenance dialysis or who received only best supportive care were excluded.” (marked-up copy; page 7, lines 8-11). In addition, we changed the relative sentences in the Abstract section (marked-up copy; page 3, lines 7, 11, 17), the Results section (marked-up copy; page 10, lines 7, 12-13, 16-18; page 13, lines 8-11, 13-14, 20, 23-24; page 14 line 8), and the Discussion section (marked-up copy; page 18, line 12).

How many patients were transplanted during follow up? I understood that eligibility for transplant was no exclusion criterion.

Reply: Thank you for your comment. Only one patient received liver transplantation (marked-up copy; page 10, lines 9-10).

Authors use the terms admission and hospitalisation – is that meant to be a surrogate for admission due to AD/ACLF? It is a bit confusing as admissions in general are not always due to AD/ACLF.

Reply: Thank you for your comment. We changed the relevant sentences (marked-up copy; page3, lines 22-23; page 8, line 9; page 13, line 9; page 14, line 6).

What is meant by curative treatment – intended to be or ultimately curative?

Reply: Thank you for your comment. That intended to be curative, so we changed curative treatment to potentially curative treatment (marked-up copy; page 10, line 8; page 11, line 11; page 12, line 21; Table 9).

How many patients developed AD/ACLF as the consequence of HCC therapy? Are there factors impacting on AD/ACLF after HCC therapy?

Reply: As shown in new Table 10, 35 cases developed AD due to HCC therapy. Three cases developed ACLF as the consequence of HCC therapy. Following your suggestion, we analyzed factors impacting on AD after HCC therapy, but we could not find the significant factor.

Were all patients with viral hepatitis treated with antivirals? Any information on viral load etc? Were there differences between treated and non-treated patients?

Reply: Thank you for your comment. We added some sentences in the Discussion section (marked-up copy; page 18, lines 4-7): 57% of patients with virus-related HCC were treated with antivirals, and there was a significant difference in the development of AD between treated and non-treated patients (treated vs. non-treated: 5.4% vs. 17.6% at 1 year/12.7% vs. 42.4% at 3 years, p < 0.001).

There are too many variables in the regression analysis. For treatment and aetiology it may be useful to have only a binary variable (categorical), e.g. treatment/no treatment or viral hepatitis/others

Reply: Thank you for your suggestion. Following your suggestion, we excluded alcohol and NASH as etiology in the regression analyses and prognostic analyses. In addition, we used potentially curative treatment in the cox regression analyses of predictive factors (marked-up copy; Table 9). 

Table 5 – Ascites, HE etc…. are no precipitating events, but complications of cirrhosis defining decompensation.

Reply: We apologize for the inappropriate term and changed “Categories of main precipitating events” to “Complications defining AD” (mar

---

## [Decision Letter · Decision Letter 1]

26 Nov 2021

PONE-D-21-23680R1Impact of acute decompensation on the prognosis of patients with hepatocellular carcinomaPLOS ONE

Dear Dr. Kondo,

Thank you for submitting your manuscript to PLOS ONE. After careful consideration, we feel that it has merit but does not fully meet PLOS ONE’s publication criteria as it currently stands. Therefore, we invite you to submit a revised version of the manuscript that addresses the points raised during the review process.

The reviewers appreciated the modifications that you made and only minor changes were requested.

We look forward to receiving your revised manuscript.

Kind regards,

Pavel Strnad

Academic Editor

PLOS ONE

Journal Requirements:

Reviewers' comments:

Reviewer's Responses to Questions

**Comments to the Author**

1. If the authors have adequately addressed your comments raised in a previous round of review and you feel that this manuscript is now acceptable for publication, you may indicate that here to bypass the “Comments to the Author” section, enter your conflict of interest statement in the “Confidential to Editor” section, and submit your "Accept" recommendation.

Reviewer #1: All comments have been addressed

Reviewer #2: All comments have been addressed

2. Is the manuscript technically sound, and do the data support the conclusions?

Reviewer #1: Yes

Reviewer #2: Yes

3. Has the statistical analysis been performed appropriately and rigorously? 

Reviewer #1: Yes

Reviewer #2: Yes

4. Have the authors made all data underlying the findings in their manuscript fully available?

Reviewer #1: Yes

Reviewer #2: Yes

5. Is the manuscript presented in an intelligible fashion and written in standard English?

Reviewer #1: Yes

Reviewer #2: Yes

6. Review Comments to the Author

Reviewer #1: The article greatly improved by the corrections made by the authors. I have still some minor comments:

- How do the authors explain that cirrhosis is not a risk factor for AD or prognosis when compared to non-cirrhotics?

- Part of the discussion is on why Albi-score and CLIF-C are associated with prognosis, but if I understand it correctly these were predictors of prognosis in the original study and not in the rebuttal. If this is the case, the authors should focus on discussion of the new predictors if they think this is needed ( ACLF, bilirubin level, HCC progression or MELD score)

- They authors suggest to perform surgical resection in patients who have HCC with preserved liver function, even at the intermediate-advanced–stage, because of the low AD incidence rate after treatment. The fact that AD is low in the resection group can also be reflected by the relative good status of patients in this group (younger age, no portal hypertension, few comorbidities). If this is the case, it is possible that the low AD occurence in the surgical group is attributed to the patient group and not to the intervention (selection bias). Did the authors compare baseline characteristics of HCC paitents in the different groups to evaluate this? If these groups are indeed different, the authors should consider to remove this conclusion.

- Please check the manuscript for spelling errors, the word cirrhosis is spelled incorrectly in some placed (for example methods section abstract)

Reviewer #2: I would like to thank the authors for their efforts to respond to all our comments. The changes applied to the manuscript improved its quality subtantially.

Authors should go through the manuscript once more to remove some remaining typos.

7. PLOS authors have the option to publish the peer review history of their article (what does this mean?). If published, this will include your full peer review and any attached files.

Reviewer #1: No

Reviewer #2: No

---

## [Author Response · Author response to Decision Letter 1]

3 Dec 2021

PONE-D-21-23680R1

Impact of acute decompensation on the prognosis of patients with hepatocellular carcinoma

PLOS ONE

To the Reviewers

Thank you for your comments and critical review.

Reviewer #1: The article greatly improved by the corrections made by the authors. I have still some minor comments:

- How do the authors explain that cirrhosis is not a risk factor for AD or prognosis when compared to non-cirrhotics?

Reply: Thank you for your comments. In our study, cirrhosis was associated with AD development (marked-up copy; page 9, lines 17-18; page 16, line 11). However, cirrhosis is not a risk factor for long-term mortality. Therefore, we added some sentences for explaining this result in the Discussion section (marked-up copy; page 15, lines 6-11). 

- Part of the discussion is on why Albi-score and CLIF-C are associated with prognosis, but if I understand it correctly these were predictors of prognosis in the original study and not in the rebuttal. If this is the case, the authors should focus on discussion of the new predictors if they think this is needed ( ACLF, bilirubin level, HCC progression or MELD score)

Reply: We agree with your suggestion. According to your suggestion, we deleted some sentences in the Discussion section (marked-up copy; page 16, lines 20-25; page 17, line 1).

- They authors suggest to perform surgical resection in patients who have HCC with preserved liver function, even at the intermediate-advanced–stage, because of the low AD incidence rate after treatment. The fact that AD is low in the resection group can also be reflected by the relative good status of patients in this group (younger age, no portal hypertension, few comorbidities). If this is the case, it is possible that the low AD occurence in the surgical group is attributed to the patient group and not to the intervention (selection bias). Did the authors compare baseline characteristics of HCC paitents in the different groups to evaluate this? If these groups are indeed different, the authors should consider to remove this conclusion.

Reply: Thank you for your comments. According to your comments, we compared baseline characteristics between the surgical and non-surgical groups. In fact, patients who received surgery had lower Child-Pugh scores than those who did not, and had no prior history of AD. Therefore, we modified some sentences in the Abstract section (marked-up copy; page 3, lines 14-15, 22-24) and the Discussion section (marked-up copy; page 14, lines 22-23; page 17, lines 10-13).

- Please check the manuscript for spelling errors, the word cirrhosis is spelled incorrectly in some placed (for example methods section abstract)

Reply: We apologize for some errors. We checked and corrected some errors in the Abstract section (marked-up copy; page 3, lines 9, 18), the Patients and Methods section (marked-up copy; page 7, line 23), the Results section (marked-up copy; page 9, line 16; page 12, lines 17-18, 25), and the Discussion section (marked-up copy; page 14, line 21).

Reviewer #2: I would like to thank the authors for their efforts to respond to all our comments. The changes applied to the manuscript improved its quality subtantially.

Authors should go through the manuscript once more to remove some remaining typos.

Reply: We would also like to thank you for taking the time to review our manuscript and apologize for some remaining typos. We checked and corrected the remaining typos in the Abstract section (marked-up copy; page 3, lines 9, 18), the Patients and Methods section (marked-up copy; page 7, line 23), the Results section (marked-up copy; page 9, line 16; page 12, lines 17-18, 25), and the Discussion section (marked-up copy; page 14, line 21).

---

## [Editor Report · Decision Letter 2]

7 Dec 2021

Impact of acute decompensation on the prognosis of patients with hepatocellular carcinoma

PONE-D-21-23680R2

Dear Dr. Kondo,

We’re pleased to inform you that your manuscript has been judged scientifically suitable for publication and will be formally accepted for publication once it meets all outstanding technical requirements.

Kind regards,

Pavel Strnad

Academic Editor

PLOS ONE

Additional Editor Comments (optional): Thank you for submitting you interesting work to PLoS One!
---

## [Editor Report · Acceptance letter]

19 Jan 2022

PONE-D-21-23680R2 

Impact of acute decompensation on the prognosis of patients with hepatocellular carcinoma 

Dear Dr. Kondo:

I'm pleased to inform you that your manuscript has been deemed suitable for publication in PLOS ONE. Congratulations! Your manuscript is now with our production department. 

Kind regards, 

on behalf of

Dr. Pavel Strnad 

Academic Editor

PLOS ONE